# Identification of Candidate Biomarker and Drug Targets for Improving Endometrial Cancer Racial Disparities

**DOI:** 10.3390/ijms23147779

**Published:** 2022-07-14

**Authors:** Pouya Javadian, Chao Xu, Virginie Sjoelund, Lindsay E. Borden, Justin Garland, Doris Mangiaracina Benbrook

**Affiliations:** 1Section of Gynecologic Oncology, Department of Obstetrics and Gynecology, Stephenson Cancer Center, University of Oklahoma Health Sciences Center, Oklahoma City, OK 73104, USA; lindsay-borden@ouhsc.edu; 2Department of Biostatistics and Epidemiology, Hudson College of Public Health, University of Oklahoma Health Sciences Center, Oklahoma City, OK 73104, USA; chao-xu@ouhsc.edu (C.X.); virginie-sjoelund@ouhsc.edu (V.S.); 3Department of Pathology, University of Oklahoma Health Sciences Center, Oklahoma City, OK 73104, USA; justin-garland@ouhsc.edu

**Keywords:** endometrial cancer, racial disparity, molecular profiling, proteomic

## Abstract

Racial disparities in incidence and survival exist for many human cancers. Racial disparities are undoubtedly multifactorial and due in part to differences in socioeconomic factors, access to care, and comorbidities. Within the U.S., fundamental causes of health inequalities, including socio-economic factors, insurance status, access to healthcare and screening and treatment biases, are issues that contribute to cancer disparities. Yet even these epidemiologic differences do not fully account for survival disparities, as for nearly every stage, grade and histologic subtype, survival among Black women is significantly lower than their White counterparts. To address this, we sought to investigate the proteomic profiling molecular features of endometrial cancer in order to detect modifiable and targetable elements of endometrial cancer in different racial groups, which could be essential for treatment planning. The majority of proteins identified to be significantly altered among the racial groups and that can be regulated by existing drugs or investigational agents are enzymes that regulate metabolism and protein synthesis. These drugs have the potential to improve the worse outcomes of endometrial cancer patients based on race.

## 1. Introduction

Endometrial cancer continues to rise in both incidence and mortality, in contrast to the decline in both statistics for most other cancers. Racial disparity is the major factor affecting endometrial cancer patient survival in the United States. Black women with endometrial cancer experience 2-fold higher mortality compared to White patients [1]. This represents one of the largest racial disparities in mortality among cancers [2]. Socioeconomic status, higher incidence of aggressive histology and comorbid conditions are known factors contributing to endometrial cancer racial disparity [3]. As these factors do not account for the entire racial disparity, it is likely that specific molecular defects and their associated pathways in the endometrial cancers also contribute [4].

A variety of cancer types, such as breast, prostate and uterine, vary clinically and pathologically based on the race of the patient. Research efforts are beginning to focus on understanding the molecular mechanisms of these differences [5]. The Cancer Genome Atlas (TCGA) analysis identified molecular subtypes of endometrial cancer associated with significantly different progression-free survival (PFS) rates [6]. These results were generated with data of endometrial cancer specimens primarily from White patients; there were insufficient Black patients to power statistical comparisons between the races. Published global profiling studies that identified mRNA and proteins that could contribute to endometrial cancer racial disparities are limited to two studies comparing endometrial cancer specimens from Black and White patients. One reported on whole transcriptome sequencing and mass spectrometry analysis of endometrial cancer samples from 17 Black and 13 White patients, which identified 89 genes that had consistent mRNA and protein differences between the two racial groups, and also concordant alterations in TCGA data of endometrial cancer patients from 49 Black and 216 White patients [7]. According to the multivariate analysis, PFS was associated with 10 of the mRNA transcripts for White patients, 9 transcripts for Black patients, and 2 transcripts for both White and Black patients. The other study reported higher transcript levels for baculoviral IAP repeat containing 7 (BIRC7), polo-like kinase 1 (PLK1) and multiple cell cycle regulatory proteins in endometrial cancer specimens from Black compared to White patients [8]. The differential molecules identified in both of these studies are involved in pathways that regulate cell cycle and cell death. In these studies, there was the limitation that the patients in the different racial groups were not matched for characteristics that may have affected the aggressiveness of their cancers or their treatment outcomes, such as age, tumor stage, weight and body mass index (BMI), although they did focus on one histology. Other studies that matched endometrial cancer tissue specimens from Black and White patients found no significant differences in gene expression patterns evaluated by microarrays. One of these studies matched the specimens from the two racial groups by stage, grade and histological subtype at various ratios [9]. The other study matched specimens from 25 Black and 25 White women for histology (endometrioid and serous), grade (1A through IVB) and stage (1–3) [10]. Each of these studies described above was limited because it compared only two races, Black and White.

In this retrospective study, we conducted a proteomic analysis comparing Black, White, American Indian and Asian groups of 10–12 endometrial cancers, each matched for age, BMI and histology. This matching was based on known involvement of these factors in endometrial cancer risk and patient outcomes. Age and obesity are well-established risk factors for increased risk and mortality of endometrial cancer [10]. Histology is known to affect patient prognosis: endometrioid is associated with more favorable outcomes, as opposed to serous, clear cell and other non-endometrioid histologies [9]. Multiple proteins were identified to be present at significantly different levels in the tumors of different races and were evaluated with bioinformatics for their potential use as biomarkers and drug targets in strategies to improve endometrial cancer patient outcomes.

## 2. Results

A total of 46 patients were included in this study (12 African Americans, 12 Whites, 12 Native Americans and 10 Asians). Table 1 summarizes the patient demographics and tumor characteristics. All samples were from patients with grade 1 endometrioid endometrial histology and stage 1. The distribution of these groups did not differ according to age; BMI; smoking status; alcohol or NSAID use; or cardiovascular or autoimmune disease. There were significantly different levels of hypertension and diabetes. The rate of diabetes was 67% in Black patients, which was significantly higher than in Whites (25%) and Asians (16%) (*p* = 0.011). The hypertension rate was significantly higher among White patients (75%) in contrast to (33%) the Asian group (*p* = 0.012).

### 2.1. Proteomic Discovery

Protein extracts from all (*n* = 46) cases were analyzed by tandem mass tag (TMT) liquid chromatography-tandem mass spectrometry (LC-MS/MS) for protein identification. As Appendix A shows, a total of 1611 proteins were identified by TMT LC-MS/MS in all endometrial samples from all groups. The gene ID listed in Appendix A is used instead of the full name for each gene in all tables of this article. The normalized data after log 2 transformation are shown in a heat map; they were analyzed using ANOVA with the adjustment of the first principal component (Appendix A). Figure 1 shows a heat map of the normalized data for the 58 proteins found to be present at significantly different levels among the races.

Some of these 58 proteins had significant differences among all four races, and a few were uniquely differentially present in one specific race compared to the other three (Figure 2 and Appendix A).

Significant differences in individual protein levels in the three other racial groups compared to the White racial group were identified for 44 proteins in the Black racial group, 24 proteins in the American Indian racial group, and 28 proteins in the Asian racial group (Table 2). Some of these protein levels were significantly different from those of the White racial group in all three races, whereas others were unique to individual Black, American Indian, or Asian racial groups.

### 2.2. Bioinformatic Analysis

The 44 proteins listed in Table 2 were evaluated by Ingenuity Pathway Analysis (IPA) to identify pathways that could be targeted for improving endometrial cancer treatments for Black, American Indian and Asian races. The top three canonical pathways identified by IPA to be the most different in endometrial cancers across racial groups (EIF2 signaling, regulation of eIF4 and p70S6K signaling and mTOR signaling) are all involved in the regulation of protein synthesis (Figure 3). These pathways were most associated with endometrial cancers from White patients; the least association was with endometrial cancers from American Indian patients. Endometrial cancers from the Asian racial group exhibited the most disparate profile of pathways compared to the other groups, due to less association with multiple pathways primarily involved in metabolism. Multiple pathways were upregulated across the races compared to the White racial group, including Coagulation System, Clathrin-mediated Endocytosis, Regulation of EIF4 and p70S, TOR Signaling and IL-12 Signaling and Production. Other pathways were upregulated in the Blacks and the Native Americans, but not in the Asians, compared to the Whites, including nNOS Signaling in Skeletal Muscle, Agrin Interactions at Neuromotor, 5 aminoimidazole Ribonulease, Citrulline-Nitric Oxide Cycle, Arginine Biosynthesis IV and Urea Cycle. 

IPA analysis of the fold differences between these proteins in each non-White group compared to the White group documented their associations with endometrial cancer (Figure 4), which provides a certain level of validation to the results of this study.

To explore the potential of the 58 identified differential proteins to be used as biomarkers and drug targets in endometrial cancer, we evaluated the differences in their mRNA and protein levels in endometrial cancers compared to healthy tissue, and their relevance to endometrial cancer patient survival using the UALCAN website (http://ualcan.path.uab.edu, accessed on 29 June 2022) [11] to probe TCGA and Clinical Proteomic Tumor Analysis Consortium CPTAC databases [11]. Table 3 shows the list of nine proteins that were found to either have an impact in endometrial cancer survival based on the TCGA database or are already targeted in cancer treatment based on literature reviews. All these proteins are expressed at significantly different levels in endometrial cancers from Black, American Indian or Asian groups compared to the White racial group. We used CPTAC data on the ULACAN website as our baseline racial data in endometrial cancer vs. normal tissue for these nine target proteins. This UALCAN website also provided a breakdown of the patient samples by race; however, the numbers of minority race patients were too low for statistical significance.

## 3. Discussion

This is the largest study to date examining the molecular profiles of stage I endometrial cancer with matched endometrioid histology, age and BMI across more than two races. Several of the proteins identified in this study have known associations with patient prognosis and survival, which supports their candidacy as biomarkers and drug targets, as described below and illustrated in Figure 5.

Two of the proteins identified in this study, ASS1 (argininosuccinate synthase 1) and PFAS (phosphoribosylformylglycinamidine synthase), regulate pyrimidine biosynthesis, which is essential for cellular growth and homeostasis maintenance. Pyrimidine biosynthesis is elevated when ASS1 is reduced or PFAS increased.

In this study, ASS1 was found to be present at significantly higher levels in endometrial cancers from patients of American Indian race, compared to White ones (mean ± SD 142.31 ± 126.13 vs. 53.22 ± 76.60 relative units (RU), *p* = 0.005). Although there is no difference in ASS1 mRNA expression between endometrial cancer samples and normal tissue in TCGA data, CPTAC data show significantly lower ASS1 expression in the primary tumor compared to normal samples, and another study found decreased ASS1 expression at the invading fronts of endometrial cancer and increased migration and invasion in endometrial cancer cells upon ASS1 knockout [26]. ASS1 is the rate-limiting enzyme in arginine biosynthesis that is deficient in multiple tumor types, resulting in decreased arginine levels [27]. Attempts to kill the tumor cells by further depleting arginine with catabolic enzymes proved unsuccessful due to development of resistance caused by upregulation of ASS1 [28]. Alternatively, an arginine-independent ASS1 effect can be taken advantage of in developing cancer therapeutics [12]. A consequence of ASS1 deficiency is accumulation aspartate, which serves a substrate and positive inducer of CAD (carbamoyl-phosphate synthase 2, aspartate transcarbamylase and dihydroorotase complex). The CAD upregulation as a consequence of ASS1 deficiency increases pyrimidine production. An alternate approach to target ASS1 deficiency that has proven effective in preclinical studies was to inhibit the upregulation of mTOR enzyme that occurs as a consequence of arginine depletion, or directly inhibiting pyrimidine synthesis using 5-fluorouracil (5-FU) [12]. Everolimus, an mTOR inhibitor, combined with letrozole, an aromatase inhibitor, is currently used for treatment of recurrent endometrial cancer based upon a positive phase 2 study [29]. 5-FU is not currently used for endometrial cancer but is widely used for other solid tumors, including first-line therapy for colorectal cancer (https://pubmed.ncbi.nlm.nih.gov/15051767/ accessed on 28 June 2022). 5-FU has been previously studied in combination with other chemotherapies for endometrial cancer but is not part of current practice (NCT00612495) [30]. Based on their significantly lower levels of ASS1, American Indian endometrial cancer patients would be expected to benefit from these approaches to deplete arginine.

PFAS, an enzyme that is directly involved in catalyzing purine salvage and de novo pyrimidine biosynthesis, was elevated in the Black (mean ± SD 152.30 ± 377.24 vs. 42.66 ± 62.94 RU, (*p* = 0.036)) and White endometrial cancer patients. CPTAC data show significantly higher PFAS protein expression in endometrial cancer compared to normal tissues. In preclinical studies, acivicin, an analog of glutamine, was shown to inhibit PFAS and other enzymes involved in purine salvage [19], resulting in growth inhibition of various cancer cell lines [20,22]. A large-scale screening study of 2000 compounds identified acivicin as a potent inhibitor of *Drosophila* tumor formation; however, RNAi knockdown studies demonstrated that this anti-tumor activity was due to inhibition of CTP synthase, another pyrimidine biosynthesis enzyme [21]. Thus, a significant amount of validation of PFAS as a drug target, and drug discovery and development research, would be needed to target PFAS in endometrial cancer. Before targeting PFAS or purine savage to improve Black endometrial cancer patient outcomes, the upregulation of ASS1 and the ultimate level of pyrimidine biosynthesis in endometrial cancer specimens would need to be determined.

Another enzyme involved in metabolism, CKB (creatine kinase B CKB), was found in this study to have higher expression in endometrial cancers in Asian patients compared to White patients (mean ± SD 755 ± 1015.9 vs. 391.39 ± 906.7 RU, *p* = 0.034). CKB had lower expression in endometrial cancer samples vs. normal tissue based on a CPTAC database of all races. CKB supports metabolism by using creatine to produce ATP. Preclinical studies demonstrated that inhibiting phosphocreatine import with the small molecule drug, RGX-202, inhibited primary, metastatic and patient derived xenograft (PDX) colorectal tumors in association with reducing phosphocreatine levels [14]. A phase 1 trial of RGX-202 in advanced gastrointestinal cancers documented increased serum and urine creatine levels consistent with the decreased creatine uptake in tumors of preclinical models treated with RGX-202 [14]. This phase 1 trial showed no dose-limiting toxicity in the first 17 patients enrolled, and a durable partial response in the highest dose cohort [13]. Combinations of various metabolic inhibitors also have potential for improving endometrial cancer outcomes based on preclinical studies. For example, RGX-202 demonstrated anti-cancer synergy with F-FU and leflunomide, an inhibitor of dihydroorotate dehydrogenase-induced nucleotide biosynthesis levels [14].

Our results showed HK2 (hexokinase 2) was elevated in endometrial cancer specimens from the Black (mean ± SD 55.31 ± 61.38 vs. 12.29 ± 9.19 RU, *p* = 0.008) and American Indian (mean ± SD 48.78 ± 64.32 vs. 12.29 ± 9.19 RU, *p* = 0.002) racial groups; the lowest expression was in the White racial group. Moreover, HK2 has higher expression in endometrial cancer vs. normal tissue based on the CPTAC database of all races. HK2 is the enzyme that catalyzes the conversion of hexoses, such as D-glucose and D-fructose, into hexose 6-phosphates (D-glucose 6-phosphate and D-fructose 6-phosphate, respectively) [31,32,33]. HK2 phosphorylation of D-glucose to D-glucose 6-phosphate is step the first of glycolysis [33]. Additionally, by preventing the release of apoptogenic molecules from the mitochondrial intermembrane space and subsequent intrinsic apoptosis, HK2 is necessary for maintaining the integrity of the outer mitochondrial membrane [34]. Various preclinical studies validate HK2 as a potential target. The long non-coding RNA (lncRNA) SNHG16 promoted the proliferation of endometrial carcinoma cells and glycolysis by competitively sponging miR-490-3p and upregulating HK2, which is miR-490-3p’s target gene [35].

Inhibition of HK2 activity with 2-deoxyglucose (2-DG) blocks the glycolysis pathway [36]. 2-DG has multiple attributes as a candidate therapeutic agent. There is substantial clinical information about 2-DG and its 2-[18F]-fluoro-2-deoxy-D-glucose (FDG) analog, as they have been extensively utilized to enhance diagnostic imaging [37]. Glucose, 2-DG and FDG have increased uptake in cancer tissues due to elevation of glucose transporters in cancer cells, which allows for the imaging of malignancies and metabolically active tissues in contrast to nonmalignant tissues [38,39]. Additionally, 2-DG is reasonably well-tolerated in cancer patients [40]. In a phase I study of oral 2-DG in patients with solid tumors, 45 mg/kg 2-DG given on a daily schedule of the first two weeks of a three-week cycle was defined as the recommended dose. This was based on cardiac QTc prolongation occurring in two of four patients at the 60 mg/kg dose level, and in none of the five patients treated at the 45 mg/kg dose level.

2-DG also has considerable potential for use in combination therapies. In preclinical studies, 2-DG sensitized cancer cells to multiple types of chemotherapeutic agent [41,42,43,44], and paclitaxel enhanced the uptake of 2-DG [45]. 2-DG sensitized pancreatic cancer cells and tumors to inhibition of MEK [46], the kinase immediately upstream of MAPK3 in the RAS-induced kinase activation cascade. Thus, combining 2-DG with conventional chemotherapies or other agents that induce glycolysis is a reasonable strategy for improving the cancer therapeutics’ efficacies without significantly increasing toxicity [47]. In a phase I trial in advanced solid tumors, daily oral 2-DG at 63 mg/kg in combination with docetaxel was chosen as the recommended phase 2 dose [48]. QTc prolongation was observed in 22% of patients, and the authors speculated that this was related to reversible hyperglycemia that occurred in 100% of the patients at the 63 to 88 mg dose levels. There was no pharmacokinetic interaction between 2-DG and docetaxel. Efficacy was not evaluated in the single-agent 2-DG trial; however, of the 34 patients in the combination trial, eleven (32%) had stable disease, one (3%) had a partial response and twenty-two (66%) had progressive disease.

Two kinases, MAPK3 and OXSR1, and a phosphatase, PTPN6, were also identified in this study to be significantly present at different levels in the races. MAPK3 (mitogen-activated protein kinase 3, also called ERK1) was present at higher levels in endometrial cancer specimens from Black compared to White patients in our study (mean ± SD 113.23 ± 45.03 vs. 27 ± 17.88 RU, *p* = 0.001); however, CPTAC data of all races demonstrated lower MAPK3 expression in endometrial cancer compared to control tissue. MAP kinases act in a signaling cascade that regulates various cellular processes, such as proliferation, differentiation, and cell cycle progression in response to a variety of extracellular signals [49,50]. BVD-523 (Ulixertinib) is a MAPK3/1 kinase inhibitor that inhibited xenograft growth in multiple cancer types and synergized with BRAF inhibitors in BRAF mutant cell lines [51]. A phase 1 study of BVD-523 in advanced solid tumors had treatment responses in subsets of patients with acceptable toxicities and pharmacokinetics [17].

OXSR1 (oxidative stress responsive 1) (mean ± SD 130.91 ± 86.07 vs. 95 ± 47.43 RU, *p* = 0.047) was highly expressed in endometrial cancer samples from Black patients; the lowest expression was in specimens from the Asian racial group. Based on CPTAC data, OXSR1 is not overexpressed in endometrial cancer compared to control tissue. OXSR1 is a serine/threonine kinase that regulates downstream kinases in response to environmental stress, and may play a role in regulating the actin cytoskeleton. The natural carotenoid lutein has been shown to reduce OXSR1 expression in oral cancer squamous cells [18].

PTPN6 (protein tyrosine phosphatase non-receptor type 6, also called Src homology region 2 domain-containing phosphatase-1 or SHP-1) was present at the highest levels in endometrial cancer specimens from Black patients and the lowest in Asian patients in this study (mean 62.72 ± 2.01 vs. 5.34 ± 0.21 RU, *p* = 0.029). In CPTAC data, PTPN6 was expressed at significantly higher levels in endometrial cancer compared to normal tissue. In endometrial cancer, PTPN6 was expressed at significantly higher levels in endometrioid in comparison to serous histology, and in association with worse prognosis in patients with endometrioid histology, but there was no association with survival of patients with serous histology [52]. Preclinical studies demonstrated that inhibition of protein tyrosine phosphatases with sodium stibogluconate inhibits hematopoietic cell line response to cytokines [53]. A phase 1 trial of sodium stibogluconate in solid tumors demonstrated safe inhibition of PTPN6 substrate dephosphorylation in peripheral blood cells [54].

Another protein identified in this study that has a related drug in development is involved in protein synthesis: EIF4A2 (eukaryotic translation initiation factor 4A2 (EIF4A2). It was elevated in endometrial cancers of the Black racial group compared to the White racial group in our study (mean 156.35 ± 163.26 vs. 41.09 ± 24.32 RU, *p* = 0.04). In CPTAC data comparing all races, there is no difference of EIF4A2 protein expression in endometrial cancer specimens compared to normal tissue. EIF4A2 is an ATP-dependent RNA helicase involved in the binding of mRNA to ribosomes [55]. Preclinical studies show that eFT266, a first in its class EIF4A inhibitor, is effective at inhibiting B-cell tumor growth in association with mTOR signaling [56]; however, its specificity for EIF4A1 versus EIF4A2 has not been robustly determined [57]. Currently, eFT266 (zotatifin) is in a phase 1 trial in combination with other drugs in patients with solid tumors (clinicaltrial.gov: NCT04092673).

The serine protease inhibitor SERPINA1 was highly expressed in endometrial cancer specimens from Asian (mean 483.88 ± 609.04 vs. 57.05 ± 52.81 RU, *p* = 0.011) and American Indian races (mean 306.92 ± 219.87 vs. 57.05 ± 52.81 RU, *p* = 0.009). The patients with the lowest expression in endometrial cancer were Black (73.36 RU) in this study. In the CPTAC dataset, SERPINA1 was significantly less expressed in endometrial cancer specimens compared to control tissues (*p* < 0.001), but had no association with patient survival. SERPINA1 is being used as an exploratory biomarker of breast cancer patient survival in a clinical trial of trastuzumab [24].

Additional proteins identified in this study warrant further investigation to determine their validity as endometrial cancer drug targets. Validation of our results with a different technology and with an independent set of specimens is needed; furthermore, the association of the biomarker with drug response should be validated before designing a clinical trial that is race-specific or race-enriched. Another limitation of our study is power limitation. With the current sample size, we had 80% power to identify a significant difference of protein level among races after multiple testing correction. There was a 0.9 SD difference among mean protein level in race groups. Furthermore, this study did not include normal tissue for comparison. The main objective was to identify differences in proteins between races that could provide biological clues to the racial disparity. Followup research using public databases was then used to determine what is already known about the identified proteins in endometrial cancer so that the proteins could be prioritized for further research as candidate biomarkers and drug targets. While Table 3 lists CPTAC data of expression of the identified proteins in endometrial cancer compared to normal tissue, it does not account for the potential that the alteration of the protein’s expression in cancer compared to normal tissue might be race-specific.

## 4. Materials and Methods

Clinically-annotated snap-frozen endometrial cancer specimens were obtained from the OUHSC Stephenson Cancer Center (SCC) Biospecimen and Tissue Pathology Shared Resource under OUHSC IRB approved protocol. Eligibility criteria for all subjects were: (a) stage I endometrial cancer; (b) matched endometrioid type histology; (c) documented age and BMI.

### 4.1. Isolation of Proteins from Endometrial Cancer Specimens

Protein was isolated from 48 matching endometrial cancer specimens from women of different races (12 Black, 12 White, 12 Native American and 12 Asian). Briefly, proteins were isolated from ~10 mg frozen endometrial cancer specimens using T-PER Tissue Protein Extraction Reagent (ThermoFisher Scientific, Waltham, MA USA). Total protein concentration was determined using bicinchoninic acid assay (BCA) reagent (Abcam; Cambridge, UK) as per manufacturer’s instruction, as previously described.

### 4.2. Mass Spectrometry Analysis

Ten micrograms of protein from each sample was subjected to the FASP protocol (Wisniewski, Methods Mol Biol (2018)) and digested with 0.2 µg Promega Sequencing Grade Modified Trypsin (Promega, Madison, WI, USA, V5111, using manufacturer’s protocols) for overnight incubation at 37 °C in 40 mM NH4HCO3. One sample containing a mix of equal amount of each sample was prepared to a total amount of 10 µg and was subjected to the same procedure as the individual samples.

After digestion, the samples were desalted, dried and resuspended in 100 mM triethylamine ammonium bicarbonate buffer. The samples were then labeled with TMT-11plex (Thermo Fischer Scientific, Waltham, MA, USA) according to manufacturer’s instructions. The TMT-11 channel was used for the “mix” sample to serve as a reference between the different TMT runs.

The TMT labeled tryptic peptides were then desalted and concentrated using Pierce™ C 18 spin columns (Thermo Fischer Scientific, Waltham, MA, USA). A total of 1 ug of tryptic peptides were loaded onto a C18 sequencing column (Acclaim^TM^ PepMap^TM^ 100 C18, ThermoFisher) and then eluted using a 120 min acetonitrile gradient for quantification. Eluted peptides were analyzed by liquid chromatography–tandem mass spectrometry (LC-MS/MS) analysis using a Thermo Lumos Fusion tribrid Orbitrap mass spectrometer coupled to an Ultimate 3000 RSLC nano ultra-high-performance liquid chromatography (UHPLC). Protein identification was performed by Proteome Discoverer version 2.4 (ThermoFisher Scientific, Waltham, MA USA) utilizing SEQUEST as the search engine and the human Uniprot reference proteome database version 20201123 (42,412 reviewed proteins). Protein identification required detection of at least two peptides per protein. Database search parameters were restricted to three missed tryptic cleavage sites, a precursor ion mass tolerance of 10 ppm, a fragment ion mass tolerance of 0.05 Da and a false discovery rate of ≤1%. Fixed protein modification was Cys carboxymethylation (+58 Da). Variable protein modifications included Met oxidation (+16 Da) and N-terminal acetylation (+42). Reporter ion intensities were bias corrected for the overlapping isotope contributions from the TMT tags according to the manufacturer’s certificate. Reporting of proteins followed the general recommended guidelines. In brief, 3 biological replicates; minimum peptide length was ≥7 amino acids; ≥2 peptide matches; protein FDR cutoff ≥ 1%. Proteins with a “Protein false discovery rate (FDR) Confidence Combined” as “High” were used for analysis; all other parameters were set as default. We corrected the total signals of each channel by computing normalization factors in order to equalize the amounts of protein labeled by each TMT reagent. In order to conduct the subsequent analysis, the data were exported to Microsoft Excel and Perseus (version 1.6.15.0, https://maxquant.net/perseus/ accessed on 28 June 2022).

### 4.3. Statistical Analysis

The proteomics data were compared between racial groups. The original data for test specimens were normalized by several steps. First, the blank values were subtracted from each sample value for each protein, and negative values were forced to zero. Then, all resulting values were multiplied by a normalization factor derived for each pool using the protein level values in the mixed control sample. Lastly, the trimmed mean of M values (TMM) was applied to remove the compositional bias [58]. The normalized data after log 2 transformation were analyzed using ANOVA with the adjustment of the first principal component, which contained the unobserved confounding effect. Further, two post-hoc multiple-comparison corrected tests for each protein having *p*-values < 0.05 in the overall ANOVA test were performed. One was Tukey’s test for pairwise comparison. The other was Dunnett’s test for the comparison between each of the minority groups and the White group. Missing data were excluded from the analysis for each protein. The significance level was 0.05. All analyses were implemented in R version 4.0 (R Core Team, 2014, Vienna, Austria) using package edgeR [59] and other base functions.

### 4.4. Bioinformatics

The proteins found to be present in endometrioid endometrial cancer specimens from women of each race at significantly different levels in comparison to at least one other set of specimens from a different race were subjected to bioinformatic analysis. The analysis goals were to validate the protein’s roles in cancer and racial disparity, and identify pathways and actionable targets in endometrial cancer specimens from patients of each race. Public databases were used, including The Cancer Genome Atlas (TCGA) and Clinical Proteomic Tumor Analysis Consortium (CPTAC), for the individual proteins. The UALCAN website (http://ualcan.path.uab.edu/index.html, accessed on 29 June 2022) was used to determine if the specific protein or its mRNA are significantly: altered in endometrial cancer compared to healthy tissue; altered in Black compared to other races; and/or are associated with survival and expression of other relevant proteins. UALCAN provides statistical analysis for the data, although in many instances the numbers of people an individual race are too low to provide sufficient power for significance. This information was carefully collated and studied for each identified gene.

Ingenuity Pathway Analysis (IPA, Qiagen) was used to gather additional information, such as the proteins’ mechanisms of action; interactions with other proteins; associations with various diseases and cellular processes; and identification of drugs that target the specific proteins. IPA utilizes published literature and existing databases to provide extensive information on individual proteins/genes and how they interact with other factors. The list of proteins found to be present at differential levels in endometrial cancer specimens from women of different races were uploaded into IPA web-based software. IPA was also used to identify how the proteins are involved in canonical pathways, cellular and disease processes and upstream/downstream molecules that can be targeted with FDA-approved or investigational agents in clinical trials.

## 5. Conclusions

In our study, specific proteins have been identified as potential biomarkers of patient prognosis and as drug targets in patients with endometrial cancer, for several races. Some of these proteins are targeted by specific drugs that are currently being, or have already been, tested in clinical trials and may be adaptable for endometrial cancer. Drug discovery efforts for regulators of the other proteins that do not already have specific drugs would be warranted upon their validation of the proteins being differentially expressed across races and functionally involved in endometrial cancer development or aggressiveness. The goal of these efforts would not be to develop a race-specific treatment, but more importantly, to improve the worse outcomes of races that experience endometrial cancer disparities.

## Figures and Tables

**Figure 1 ijms-23-07779-f001:**
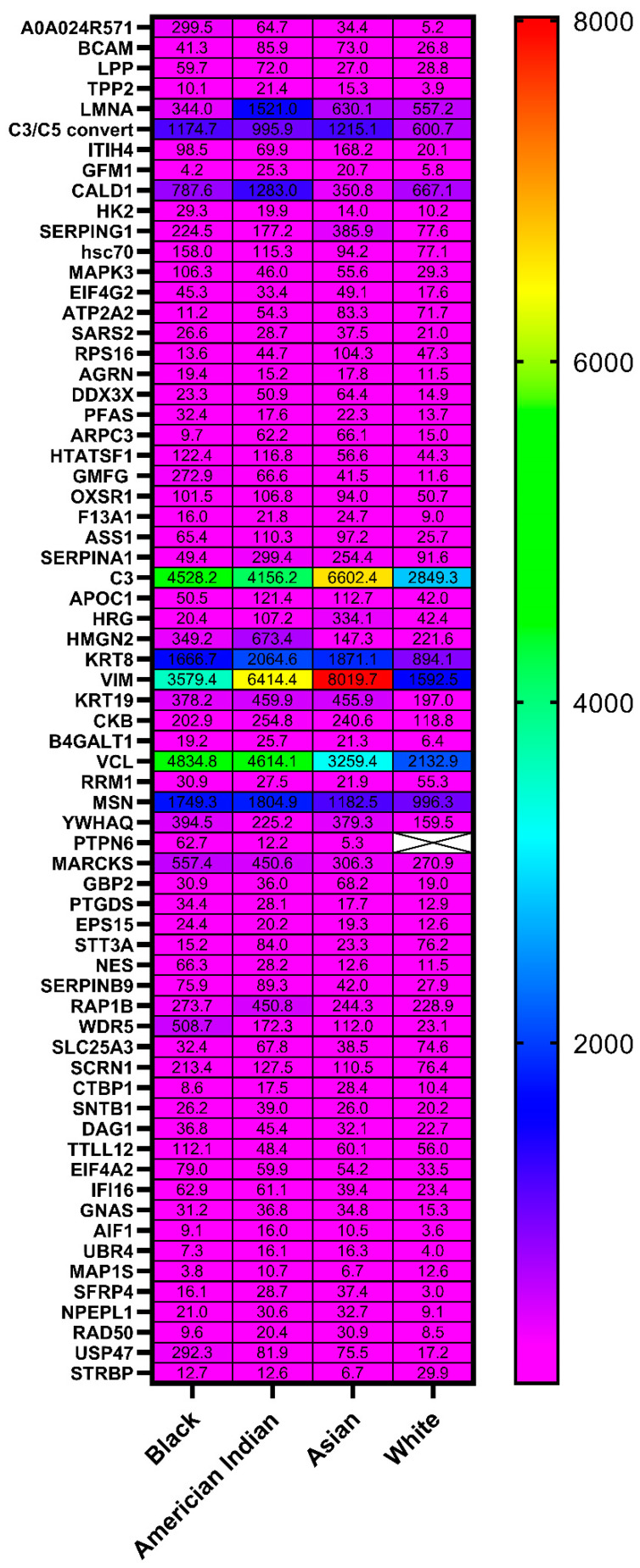
Heat map of log2 normalized levels for each protein found to be present at significantly different levels across the four races.

**Figure 2 ijms-23-07779-f002:**
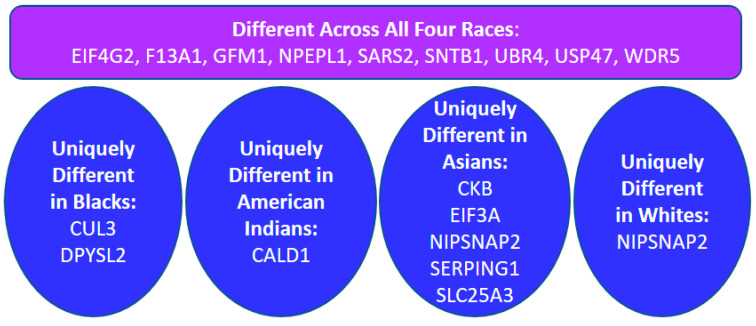
Comparison of protein differences across all racial groups.

**Figure 3 ijms-23-07779-f003:**
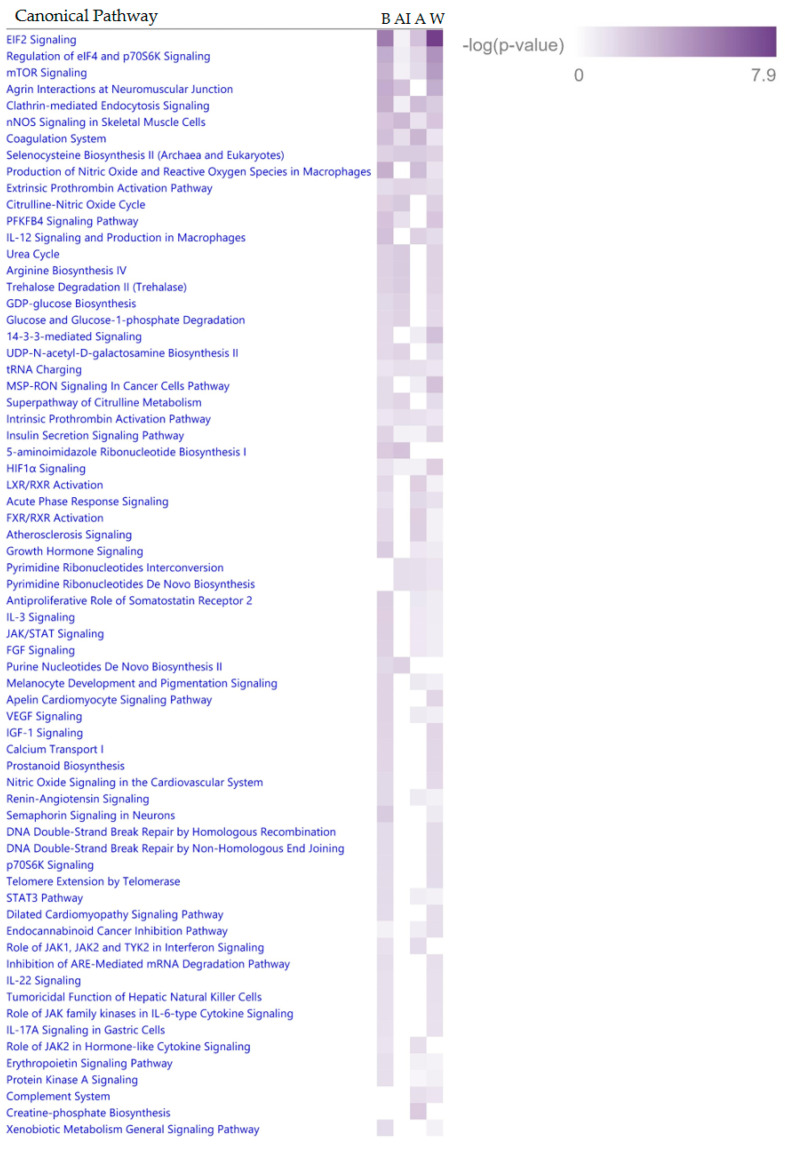
IPA-identified canonical pathways associated with 44 proteins present at significantly different levels in endometrial cancer specimens from Black, African American or Asian compared to the White race. B: Black, AI: American Indian, A: Asian, W: White.

**Figure 4 ijms-23-07779-f004:**
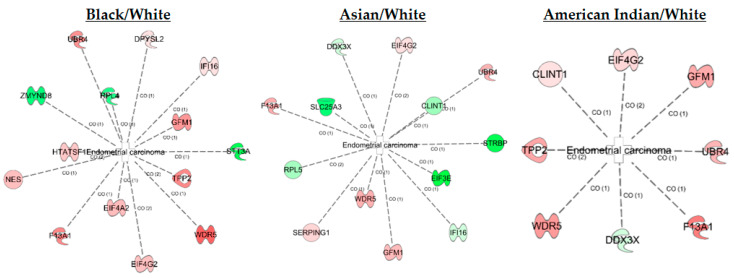
Association of differential proteins with endometrial cancer. These pathways provide a visualization of fold change increase (red) or decrease (green) in the specific racial group compared to the White racial group.

**Figure 5 ijms-23-07779-f005:**
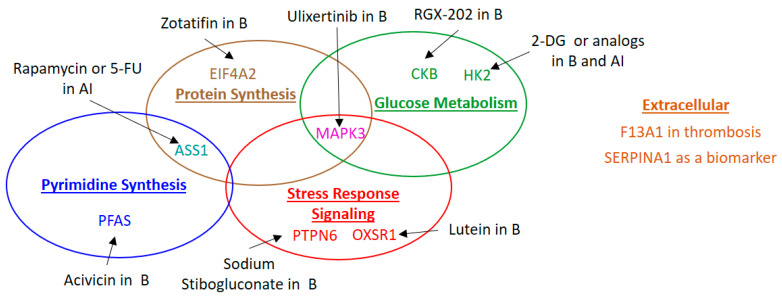
Illustration of functions and available drugs or drug candidates for identified targetable proteins.

**Table 1 ijms-23-07779-t001:** Comparison of patient characteristics between groups.

	Black (*n*:12)	White (*n*:12)	American Indian (*n*:12)	Asian (*n*:10)	*p*
Age (yrs)	63.6 ± 13.5	59.2 ± 6.2	58.1 ± 10.5	64.1 ± 12.5	0.45
BMI	39.2 ± 10.3	35.4 ± 7.1	42.2 ± 8.7	38.6 ± 8.4	0.27
Smoking	2	7	2	1	0.29
Hypertension	5	9	7	4	0.012
Alcohol use	1	2	1	0	0.84
NSAID use	1	0	1	0	0.15
Cardiovascular disease	3	3	5	2	0.21
Autoimmune disease	1	0	0	0	0.44
Diabetes	8	3	7	2	0.011

**Table 2 ijms-23-07779-t002:** List of gene IDs for proteins significantly different in endometrial cancer specimens from Black, American Indian and Asian racial groups compared to the White racial group.

Race	Gene IDs	
	Higher Concentration	Lower Concentration
**Black**	AIF1, AGRN, ASS1, CUL3, DAG1, DPYSL2, EHD1, EIF4A2, EIF4G2, EPS15, F13A1, GMFG, GFM1, HK2, HTATSF1 IFI16, MAPK3, NES, NPEPL1, OXSR1, PTPN6, PTGDS, PFAS, RAB5B, RAD50, SCRN1, SNX1, SNTB1, SERPINB9, SARS2, TPP2, UBR4, USP47, WDR5, YWHAQ,	ATP2A2, APOC1, MAP1S, RPL4, RPL23, RPS16, SERPINA1, STT3A, ZMYND8
**American Indian**	AGRN, ASS1, AIF1, CLINT1, CALD1, DAG1, EIF4G2, EPS15, F13A1, GFM1, HK2, HMGN2, KRT19, NPEPL1, SNX1, SNTB1, SARS2, UBR4, USP47, TPP2, WDR5	DX3X, MAP1S, PFAS
**Asian**	APOC1, CKB, EIF4G2, F13A1, GBP2, GFM1, HMGN2, KRT19, NPEPL1, RAB5B, SNTB1, SERPING1, SARS2, SERPINA1, UBR4, USP47, VIM, WDR5	CLINT1, DDX3X, EIF3E, GBAS, IFI16, PTPN6, OXSR1, RPL5, SLC25A3, STRBP

**Table 3 ijms-23-07779-t003:** Targetable differential proteins that have reported survival and prognostic value.

Gene Symbol	Expression in Endometrial Cancer vs. Normal Tissue ¥ [11]	Biomarker-Driven Therapy	Disease or Use	Clinical Trial Phase	Impact on Endometrial Cancer Patient Survival (TCGA) † [11]	Expression in Our Study Cohort
**ASS1**	Lower (*p* < 0.0001)	Rapamycin/ mTOR inhibtors, or fluorouracil (5-FU) [12]	Multiple cancers	Rapamycin Approved/Phase 2	Worse outcome with higher expression	AI (High) W (Low)
**CKB**	Lower (*p* < 0.0001)	RGX-202 [13,14]	Gastrointestinal cancer	Phase 1	No	A (High) W (low)
**EIF4A2**	No significant difference	Zotatifin [15]	Solid tumors	Phase 1–2	Worse outcome with higher expression	B (High) W (Low)
**HK2**	Higher (*p* < 0.0001)	2-DG and analogs [16]	Prostate cancer, PET imaging	Phase 2	No	B and AI (High) W (Low)
**MAPK3**	Lower (*p* < 0.0001)	Ulixertinib [17]	Solid tumors	Phase 2	No	B (High) W (Low)
**OXSR1**	No significant difference	Lutein [18]	Oral cancer	Preclinical	No	B (High) A (Low)
**PFAS**	Higher (*p* = 0.0002)	Acivicin [19,20,21,22]	Liver cancer	Preclinical	No	B (High) AI (Low)
**PTPN6**	Higher (*p* < 0.0001)	Sodium stibogluconate [23]	Melanoma	Phase 1	Worse outcome with higher expression	B (High) A (Low)
**SERPINA1**	Lower (*p* < 0.0001)	Trastuzumab [24].	Breast cancer	Phase 2 (Exploratory Biomarker)	No	A and AI (High) B (low)

B: Black, AI: American Indian, A: Asian, W: White—with the highest expression indicated by (High) and the lowest expression indicated by (Low); PET: positiron emission tomography, **¥** based on proteomic expression in the CPTAC dataset [11,25], † survival based on the TCGA dataset [11,25].

## Data Availability

Data results for the mass spectrometry experiments are provided in the Appendix A.

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
