# Peer review of "Identification of Candidate Biomarker and Drug Targets for Improving Endometrial Cancer Racial Disparities"

_ijms, 2022, doi:10.3390/ijms23147779_

Round 1

Reviewer 1 Report

In the manuscript "Identification of candidate biomarker and drug targets for proving endometrial cancer racial disparities", the authors described the potential as biomarkers of different proteins across endometrial cancer based on racial groups.

The introduction seems well supported and the methodological approach appropriate. However in the conclusion, the authors need to highlight that the sample size in each group of patients is very low, and the proteomic data need further validation.

In the manuscript some improvements can be done:

1) increase the quality/ resolution of figures to allow a good reading.

2) the nomenclature to identify the different patient groups across the manuscript should be consistent. One good option and as used in table 3 could be the "B: Black, AI:American Indian, A: Asian, W: White". The group designation should be consistent in all manuscript lengths.

3) In the discussion, the introduction of a figure with signaling pathways of the most important proteins identified and where the current drugs act will be helpful in understanding the possible biological effects of using the new drugs described.

4) the authors must read carefully the manuscript to correct some typing errors as F-Fu instead of 5-FU, among others.

Author Response

Referee 1.

In the manuscript "Identification of candidate biomarker and drug targets for proving endometrial cancer racial disparities", the authors described the potential as biomarkers of different proteins across endometrial cancer based on racial groups.

The introduction seems well supported and the methodological approach appropriate.

We thank the reviewer for these positive comments.

However in the conclusion, the authors need to highlight that the sample size in each group of patients is very low, and the proteomic data need further validation.

            We described these limitations in the final paragraph of the discussion section, because we believe that they are points of discussion rather than conclusions of the study.

In the manuscript some improvements can be done:

1) increase the quality/ resolution of figures to allow a good reading.

We are providing the figures as Tiff files in addition to inserting them in the Word document.

2) the nomenclature to identify the different patient groups across the manuscript should be consistent. One good option and as used in table 3 could be the "B: Black, AI:American Indian, A: Asian, W: White". The group designation should be consistent in all manuscript lengths.

            We have carefully gone through and corrected the terms to have the full races listed in the text and the abbreviations consistently used across all figures and tables.

3) In the discussion, the introduction of a figure with signaling pathways of the most important proteins identified and where the current drugs act will be helpful in understanding the possible biological effects of using the new drugs described.

            Figure 5 was created and used to address this concern.

4) the authors must read carefully the manuscript to correct some typing errors as F-Fu instead of 5-FU, among others.

              We have carefully gone through the manuscript to correct errors and assure consistency in nomenclature.

Reviewer 2 Report

Finding predisposing factors for malignant disorders is very important issue. The authors present a proteomic study which analyzed 48 matching endometrial cancer specimens by MS. Although the method used is very sophisticated and laborous, I think the conclusions that could be drawed are very limited at this point. 

First of all, they didnt (or couldn't) use any control samples. So it can not really be deduced whether the racial protein expressional differences are related to endometrial cancer or not? In other words, how representative are the differences for the races in general? Do we know whether the selected proteins increase or decrease from baseline - in all 4 races- in endometrial cancer? (Comparing data to TCGA only partially solves the lack of baseline data issue) 

Another concern is the relatively small sample size, given that large individual differences could be observed across all proteins. Speaking of which, I think statistical parameters such as SD and p values should also be presented in the main text/figures, not just in the Supplementary file. 

Minor points: 

- line 259: please check the grammar. Seems to be an unfinished sentence.

- Discussion is way too long, I think it is too early at this point to draw therapeutic conclusions on individual proteins at n=<12.

- I'd suggest to choose the average value of one of the races instead of a mixed sample, would be easier to compare the differences between races.    

Author Response

Referee 2.

Finding predisposing factors for malignant disorders is very important issue. The authors present a proteomic study which analyzed 48 matching endometrial cancer specimens by MS. Although the method used is very sophisticated and laborous, I think the conclusions that could be drawed are very limited at this point. 

First of all, they didnt (or couldn't) use any control samples. So it can not really be deduced whether the racial protein expressional differences are related to endometrial cancer or not? In other words, how representative are the differences for the races in general? Do we know whether the selected proteins increase or decrease from baseline - in all 4 races- in endometrial cancer? (Comparing data to TCGA only partially solves the lack of baseline data issue) 

We addressed this issue by evaluating TCGA data for expression of the identified proteins in endometrial cancer compared to normal tissue and listed this in Table 3.  We also added this concern to the final paragraph of the discussion where study limitations are described.

Another concern is the relatively small sample size, given that large individual differences could be observed across all proteins. Speaking of which, I think statistical parameters such as SD and p values should also be presented in the main text/figures, not just in the Supplementary file. 

            In addition to describing this limitation in the last paragraph in the discussion section, we now have listed the statistical parameters in the main text.

Minor points: 

- line 259: please check the grammar. Seems to be an unfinished sentence.

            We apologize for this issue and have adjusted the text.

- Discussion is way too long, I think it is too early at this point to draw therapeutic conclusions on individual proteins at n=<12.

            We believe that our wording in the discussion that the known associations of the “identified” proteins supports their “candidacy as biomarkers and drug targets” is an honest presentation of our research. We believe that the discussion provides readers details they will need in considering whether or not the “candidate biomarkers and drug targets” are worth pursuing.  Furthermore, the details in the discussion can be used to support the rationale and design of future studies.

- I'd suggest to choose the average value of one of the races instead of a mixed sample, would be easier to compare the differences between races.    

In this manuscript, we evaluated the data in two ways.  First we compared the values across all as shown in Figure 1.  Then we compared the White race to the other three races as shown in Table 2 and Figure 4 to identify potential candidates that could be targeted to improve endometrial cancer racial disparities.

Round 2

Reviewer 2 Report

I think that my two main concern are still not addressed properly. Lack of racial baseline expressional data makes the validity of the whole study questionable. Simply stating my concerns as a limitation in the discussion did not solve the issue satisfactory. 

A suggestion: ualcan.path.uab.edu has satisfactory amount of 'normal' vs. caucasian cancer data. I would use that as a reference point and evaluate the racial differences found accordingly.  

Regarding the TCGA, it is not clear for me whether the authors compare their data to the mRNA or the protein dataset? Please clearly indicate in Table 3 and throughout the text whether you used protein or mRNA levels from the TCGA database! (Obviously protein expressional levels would be preferred over mRNA data)

Secondly, I would rather prefer that table 2 (which lists the 58 proteins that the author claim to be significant) would contain the SD and p values, but I guess mentioning them at the 9 selected proteins is sort of OK. But then, this should be in the results section and not in the discussion. (Again, discussion is way too long anyway.)  

Author Response

I think that my two main concern are still not addressed properly. Lack of racial baseline expressional data makes the validity of the whole study questionable. Simply stating my concerns as a limitation in the discussion did not solve the issue satisfactory. 

A suggestion: ualcan.path.uab.edu has satisfactory amount of 'normal' vs. Caucasian cancer data. I would use that as a reference point and evaluate the racial differences found accordingly.  

Thank you very much for this valuable comment, we agree with the importance of referencing cancer compared to normal tissue and refer the reviewer to the Table 3 second column title “Expression in endometrial cancer vs. normal tissue” this is based on ualcan.path.uab.edu CPTAC data as the respectable reviewer suggested. We have added the reference and footnote to elaborate this point in the text as well. In the updated result section there are multiple sentences describing CPTAC protein baseline data as reference point for normal tissue.

Regarding the TCGA, it is not clear for me whether the authors compare their data to the mRNA or the protein dataset? Please clearly indicate in Table 3 and throughout the text whether you used protein or mRNA levels from the TCGA database! (Obviously protein expressional levels would be preferred over mRNA data)

We have updated table 3 and text accordingly based on the reviewer’s comment. We agree with the reviewer that the protein expression is preferred and used the CPTAC protein dataset to evaluate differences in expression between normal vs. endometrial cancer sample. Because the CPTAC database does not have patient survival information, our survival data is absed on the TCGA mRNA dataset. This is now clarified in the table reference and footnote.

Secondly, I would rather prefer that table 2 (which lists the 58 proteins that the author claim to be significant) would contain the SD and p values, but I guess mentioning them at the 9 selected proteins is sort of OK. But then, this should be in the results section and not in the discussion. (Again, discussion is way too long anyway.)  

We have added a new sheet (sheet 4) to the supplement S1 summarizing descriptive information for all 58 proteins which includes Mean and SD, sheet 3 has all information regarding P value for all 58 genes. We initially transferred all the descriptive information to the results section and tried to make the discussion part shorter as it was suggested by the reviewer, however this made the article disconnected and difficult to read.  Thus, we request the reviewers consideration in allowing us to retain the current discussion format which refers back to the results but does not reiterate the results.  All of the results are presented in the tables in the results section and the discussion highlights their importance relevant to what is known about the proteins. Again, we tried to give a focused view based on “identified” proteins which supports their “candidacy as biomarkers and drug targets”. We believe that the discussion provides readers details they will need in considering whether or not the “candidate biomarkers and drug targets” are worth pursuing. 

Round 3

Reviewer 2 Report

"Thank you very much for this valuable comment, we agree with the importance of referencing cancer compared to normal tissue and refer the reviewer to the Table 3 second column title “Expression in endometrial cancer vs. normal tissue” this is based on ualcan.path.uab.edu CPTAC data as the respectable reviewer suggested. We have added the reference and footnote to elaborate this point in the text as well. In the updated result section there are multiple sentences describing CPTAC protein baseline data as reference point for normal tissue." 

This is still not what I asked for....

CPTAC data tells you what is the difference between normal and the cancer of the caucasian race. Granted, normal has an unknown race composition, but it is at least a reference point you can start from. You should correspond your caucasian race protein expressional level data to the CPTAC caucasian cancer values so you could also compare your other races to the CPTAC normal levels. 

In other words; if CPTAC tells you that the protein elevated 5X in caucasian ovarian cancer, how much did the other races elevate, based on your data? 

Author Response

We appreciate the reviewer's clarification and now better understand the concern.  Unfortunately, we cannot know the level of individual protein upregulation in cancer over normal in our data, because our experiment did not include normal tissue.  The major objective of the study was to compare levels of proteins in endometrial cancer between races in order to identify differences, and then evaluate what is known about those proteins based on public data (including fold elevation in cancer over normal) and their potential to be used as drug targets to improve endometrial cancer patient outcomes in races that have disparity, with potential for application to all races.  Also, we cannot compare exact protein level expression with the CPTAC data due to differences in technologies used.